# Does a Higher Protein Diet Promote Satiety and Weight Loss Independent of Carbohydrate Content? An 8-Week Low-Energy Diet (LED) Intervention

**DOI:** 10.3390/nu14030538

**Published:** 2022-01-26

**Authors:** Jia Jiet Lim, Yutong Liu, Louise Weiwei Lu, Daniel Barnett, Ivana R. Sequeira, Sally D. Poppitt

**Affiliations:** 1Human Nutrition Unit, School of Biological Sciences, University of Auckland, Auckland 1024, New Zealand; yliu561@aucklanduni.ac.nz (Y.L.); louise.lu@auckland.ac.nz (L.W.L.); i.sequeira@auckland.ac.nz (I.R.S.); s.poppitt@auckland.ac.nz (S.D.P.); 2Riddet Institute, Palmerston North 4474, New Zealand; 3Department of Medicine, University of Auckland, Auckland 1010, New Zealand; 4High-Value Nutrition National Science Challenge, Auckland 1023, New Zealand; 5Department of Statistics, University of Auckland, Auckland 1010, New Zealand; daniel.barnett@auckland.ac.nz

**Keywords:** low-energy diet, partial diet replacement, body composition, energy intake, appetite regulation, gut peptides, amino acids

## Abstract

Both higher protein (HP) and lower carbohydrate (LC) diets may promote satiety and enhance body weight (BW) loss. This study investigated whether HP can promote these outcomes independent of carbohydrate (CHO) content. 121 women with obesity (BW: 95.1 ± 13.0 kg, BMI: 35.4 ± 3.9 kg/m^2^) were randomised to either HP (1.2 g/kg BW) or normal protein (NP, 0.8 g/kg BW) diets, in combination with either LC (28 en%) or normal CHO (NC, 40 en%) diets. A low-energy diet partial diet replacement (LED_pdr_) regime was used for 8 weeks, where participants consumed fixed-energy meal replacements plus one *ad libitum* meal daily. Four-day dietary records showed that daily energy intake (EI) was similar between groups (*p* = 0.744), but the difference in protein and CHO between groups was lower than expected. Following multiple imputation (completion rate 77%), decrease in mean BW, fat mass (FM) and fat-free mass (FFM) at Week 8 in all was 7.5 ± 0.7 kg (*p* < 0.001), 5.7 ± 0.5 kg (*p* < 0.001), and 1.4 ± 0.7 kg (*p* = 0.054) respectively, but with no significant difference between diet groups. LC (CHO×Week, *p* < 0.05), but not HP, significantly promoted postprandial satiety during a preload challenge. Improvements in blood biomarkers were unrelated to LED_pdr_ macronutrient composition. In conclusion, HP did not promote satiety and BW loss compared to NP LED_pdr_, irrespective of CHO content.

## 1. Introduction

Energy-restricted diets, in particular commercial Low-Energy Diets (LED), have been shown to be a successful strategy to promote body weight (BW) loss in overweight and obesity [1,2]. However, the optimal macronutrient composition for successful BW loss remains unresolved [3,4,5]. Whilst a higher protein (HP) intake is hypothesised to promote BW loss [6,7,8,9], it is often confounded by the accompanying lower carbohydrate (CHO) content [10]. To provide substantiation for protein-induced BW loss, the European Food Safety Authority [11] deemed it necessary to untangle the effect of dietary protein from CHO.

The recommended dietary intake (RDI) for protein in adults is 0.8 g/kg BW [12]. Whilst this RDI is designed to maintain nitrogen balance during energy-balance, increasing protein intake to 1.2 g/kg BW has been proposed to improve BW loss outcomes [13,14], due in part to beneficial effects reported for appetite regulation [15], retention of fat-free mass (FFM) [16], and enhanced diet-induced thermogenesis [6,17]. The Institute of Medicine also developed the Acceptable Macronutrient Distribution Range (AMDR), recommending 45–65% CHO, 15–25% protein, and 20–30% fat by energy at energy balance to promote adequate intake of various nutrients [12]. Yet, it has long been established that dietary CHO is not nutritionally essential for body function [18], which becomes part of the foundation for proponents of very-low CHO diets. Substantial restriction of dietary CHO intake to <50 g/day as in a ketogenic diet has been shown to promote BW loss [19,20,21], with some evidence of increased satiety [22]. Hence, in dietary studies where HP (1.2 g/kg BW) versus normal (0.8 g/kg BW) levels of protein intake have been compared, it has been difficult to attribute changes in BW to HP *per se* due to concurrent iso-energetic substitution of other macronutrients, such as CHO. We are aware of only 1 prior trial, a carefully designed multi-factorial isoenergetic protein/CHO intervention, where Soenen et al. concluded HP to be causal in successful BW loss, and not the lower CHO component of the diet [10].

Despite this potential for HP to promote BW loss, commercial total diet replacement (TDR) LED formulations provide only adequate protein to meet the RDI [23]. Using an LED as a primary driver for negative energy balance, we were interested in testing the effect of HP on satiety, BW loss and the retention of FFM within this LED paradigm. To allow for both the satiating and thermic effects of protein to be translated into a clinically meaningful difference in energy balance and hence BW, it was essential to include an *ad lib* meal component in the diet [24], which was done through a partial diet replacement (PDR) regime [25]. PDRs differ from TDRs as the former comprises multiple fixed energy meal replacements (i.e., shake, soup, bar, or porridge sachets) plus one variable energy food-based meal each day (i.e., home-cooked evening meal). *Ad lib* consumption to appetite of the home-made meal allows the purported satiating effect of protein to manifest into variable EI, hence variable BW loss.

One of the mechanisms by which HP diets are hypothesised to promote satiety is through secretion of glucagon-like peptide-1 (GLP-1) and peptide YY (PYY) from the nutrient-sensing enteroendocrine cells [26]. GLP-1 and PYY are commonly purported as “satiety” peptides, albeit there may be a threshold concentration that must be achieved through dietary intervention to induce a physiological appetite response [27,28]. In acute postprandial studies, higher concentration of some plasma amino acids (AAs) has also been implicated in appetite regulation, such as branched-chain AAs [29] and glycine [30,31,32]. However, branched-chain AAs and aromatic AAs have been shown to predict the development of type-2 diabetes [33,34]. Therefore, using HP diets for promoting BW loss may also require careful assessment of the effects on plasma AAs.

This intervention is further built upon Soenen’s study [10], aiming to evaluate whether HP can induce successful BW loss independent of concurrent lower-CHO (LC) content, under conditions of negative energy balance achieved using an 8-week LED partial meal replacement (LED_pdr_) regime. We hypothesised that, in adults with obesity, an HP LED would promote greater satiety and BW loss, when compared to a standard LED meeting the protein RDI, independent of LC.

## 2. Materials and Methods

### 2.1. Trial Design

This study was an unblinded, randomised 2 × 2 factorial, 8-week, parallel design trial, comparing the effect of HP (1.2 g/kg BW, 50%en) versus normal protein (NP, 0.8 g/kg BW, 35%en) LED, and LC (28%en) versus normal CHO (NC, 40%en) LED on BW loss. Dietary fat constituted the remaining energy in the LED for all dietary groups. LED intervention commenced on the day after baseline assessment at Week 0 through to Week 8, whereas post-intervention assessment was conducted on the day after intervention completion. The trial was conducted at the Human Nutrition Unit (HNU), University of Auckland, New Zealand, between March and December 2019. The trial complied with the Good Clinical Practice, received ethical approval from Auckland Human Disability Ethics Committee (Reference: 18/CEN/238) and was prospectively registered with the Australia New Zealand Clinical Trial Registry (Reference: ACTRN12619000209190). Participants received a Participant Information Sheet and provided written informed consent before data collection.

### 2.2. Participants

Advertisements were posted on social media platforms and public notice boards in Auckland, New Zealand. Previous HNU participants were also invited. Prospective participants were pre-screened via an online questionnaire or a telephone interview, then invited to the HNU for an on-site screening visit. The inclusion criteria were (i) females aged 18–65 years, (ii) and body mass index (BMI) 30–45 kg/m^2^ with a maximum BW of 130 kg. Exclusion criteria were: (i) BW change >5% in the previous 3 months; (ii) taking part in an active diet program; (iii) using medications or suffering from conditions known to affect BW and/or appetite; (iv) prior bariatric surgery; (v) impaired liver or kidney function; (vi) significant current disease, such as stage 2 hypertension, type-2 diabetes, cardiovascular disease, cancer, or digestive disease; (vii) depression or anxiety; (viii) unable or unwilling to consume food items included in the study; (ix) smokers or ex-smokers ≤ 6 months; (x) pregnant or breastfeeding; or (xi) unwilling or unable to comply with the study protocol.

### 2.3. LED Intervention

The LED intervention was delivered following a PDR protocol. The daily EI was approximately 40% of the daily energy requirement, estimated using the following equation:40% of estimated daily energy requirement= 0.4 × BMR (Harris-Benedict’s Equation for women) × Estimated Physical Activity Level (PAL)= 0.4 × [655 + (9.6 × weight in kg) + (1.8 × height in cm) − (4.7 × age in years)] × 4.184 (conversion from kcal to kJ) × 1.375 (assumed undertaking light activity at work)(1)

Based on the calculated energy requirement, participants were assigned to the 3.9 MJ or the 4.6 MJ diet plan. The diet plan was designed at two levels of energy to ensure participants with different energy requirement were at similar level of energy deficit. Participants were instructed to consume 4 meals daily, consisting a participant-homemade oatmeal porridge supplemented with whey protein (Nutra Whey Natural, Nutratech Ltd., Papamoa, New Zealand) and psyllium husk as breakfast, 2 sachets of meal replacements (Cambridge Weight Plan, Auckland, New Zealand) as lunch and afternoon snack, plus a participant-homemade variable meal as dinner. This PDR format is summarised as the “1 + 2 + 1” format (Figure 1). Participants were provided with the whey protein and meal replacements at no cost, but were required to purchase other ingredients for the oatmeal porridge and the dinner meal. They received a recipe booklet specific to their randomisation, a home kitchen scale for weighing food ingredients, and instructions on meal preparations using pre-recorded videos. 

The oatmeal porridge and meal replacements each provided fixed energy (0.8 MJ), collectively known as the fixed meals, and were required to be consumed in their entirety. In contrast, the energy allowance of the variable meal was either 1.6 MJ for the 3.9 MJ diet plan, or 2.2 MJ for the 4.6 MJ diet plan. Participants were requested to consume the variable meal until comfortably full (*ad lib*). The macronutrient composition planned for each diet group is summarised in Table 1.

Apart from the differences in macronutrient composition, all participants were given standard dietary advice to: (i) drink at least 2.25 L water daily; (ii) consume zero-energy sweetened or flavoured beverages, jelly, and sweets in moderation to curb cravings; (iii) avoid drinking energy-containing beverages; (iv) avoid drinking more than one serving of caffeinated beverage daily; (v) avoid eating additional foods, including fruits and vegetables, not included in the diet plan. To promote compliance with the study protocol, participants attended fortnightly dietary consultation meetings at the HNU over the 8-week period with a registered dietitian, in their randomised groups. The dietary advice conformed with the Clinical Guidelines for Weight Management in New Zealand [35]. Participants were encouraged to continue their habitual daily activities during the study, but were asked to refrain from vigorous physical activity, as part of the standard LED protocol [36].

### 2.4. Dietary Intake Assessment

Energy and macronutrient intake was assessed via 4-day weighed dietary records, in addition to 24-h nitrogen balance through the collection of a 24-h urine sample, at Week 0, 4 and 8. Dietary records were analysed using Foodworks (Xyris 8.0 Professional, Brisbane, Australia), checked independently by a second research staff member. Participants were given a urine collection kit to collect 24-h urine samples for the measurement of urine urea, and were interviewed to assess the completeness of 24-h urine collection. Protein intake was estimated from urine urea using the following equation [37]:(2)$$\mathrm{Estimated}\;\mathrm{protein}\;\mathrm{intake}\;\left(g\right)=\frac{24-\mathrm{hour}\;\mathrm{urea}\;\mathrm{nitrogen}\;\left(g\right)+\left(0.031\;\times \;\mathrm{BW}\right)}{0.16}.$$

Protein intake estimated from dietary records and urine urea were then compared using Bland–Altman analysis [38]. Due to an *ad lib* component of this dietary intervention, it was expected that total energy and macronutrient intake would vary, reflecting the appetite changes. The 24-h urine sample was collected to determine the accuracy of dietary records. 

### 2.5. Anthropometry and Body Composition Measurements

The primary outcome of this study was BW measured at Week 0, 4, and 8. All participants were weighed in light clothing after an overnight fast. Height was measured at the Screening Visit, whereas waist circumference, waist-hip ratio (WHR), systolic blood pressure (SBP) and diastolic blood pressure (DBP) were measured at Week 0, 4 and 8. Body mass index (BMI) was also calculated. Fat mass (FM) and fat-free mass (FFM) were measured using dual-energy X-ray absorptiometry (DXA, iDXA software version 15, GE-Lunar, Madison, WI, USA) at Week 0 and 8. Percentage FM was calculated as FM × 100/(FM + FFM). The timing of the body composition scan was standardised with the measurement conducted in the afternoon (1300–1430 h) after the preload postprandial challenge protocol (see Section 2.7).

### 2.6. Blood and Urine Samples

Fasted venous blood samples Week 0, 4 and 8. Blood samples were collected into a BD^TM^ Vacutainer containing fluoride oxidase for the measurement of plasma glucose; BD^TM^ Vacutainer containing dipotassium ethylenediaminetetraacetic acid (K_2_EDTA) for the measurement of plasma amino acids (AAs); BD^TM^ Serum Separator Tube II Advance Vacutainer for the measurement of serum lipids, including total cholesterol (TC), high-density lipoprotein cholesterol (HDL-C), low-density lipoprotein cholesterol (LDL-C), triglyceride (TG), and non-esterified fatty acid (NEFA); BD^TM^ P800 Vacutainers containing proprietary cocktails of peptide inhibitors for the measurement of plasma insulin, glucagon, gastric-inhibitory polypeptide (GIP), glucagon-like peptide-1 (GLP-1) and peptide YY (PYY). Plasma and serum samples were obtained by centrifugation at 1500× *g* for 10 min at 4 °C. Urine sample collection was described in Section 2.6. Aliquots of plasma, serum, and urine were stored at −80 °C until batch analysis.

Plasma glucose, serum lipids and urine urea were measured using a Cobas^®^ c311 analyser (Roche, Mannheim, Germany). Plasma insulin, glucagon, GIP, GLP-1 and PYY were measured using MILLIPLEX^®^ MAP Human Metabolic Hormone Magnetic Bead Panel 96-Well Plate Assay (HMHEMAG-34K, Merck Millipore, Germany). Plasma AAs were measured using Ultra-High-Performance Liquid Chromatography assay with pre-column derivatisation using AccQ-Tag [39,40].

### 2.7. Appetite Assessment—Preload Postprandial Challenge Protocol

The subjective feelings of appetite were assessed onsite at HNU after a 10–14 h fast at Week 0 and Week 8, measured using Visual Analogue Scale (VAS). Participants received a 1.8 MJ standardised mixed meal breakfast (27%en protein, 33%en fat, 37%en available CHO, and 3%en fibre), consisting of toast with peanut butter, a hard-boiled egg, a chicken and mushroom meal replacement soup (Cambridge Weight Plan, Auckland, New Zealand) and 250 mL water at 0900 h. The energy content of the standardised mixed meal was designed to match the median energy content of preloads previously identified in our review [28], whereas the higher protein content (27%en, 30 g) was conjectured to augment the postprandial satiety response and to facilitate the detection of changes in postprandial satiety following the LED intervention. Participants were required to consume the breakfast in its entirety within 15 min. VAS was administered before breakfast (t = 0 min), immediately after breakfast at t = 15 min, and subsequently at t = 30, 60, 90, 120, 150, 180, and 210 min. One-third of participants voluntarily underwent repeated blood samplings at time intervals concurrent to the appetite VAS for the measurement of postprandial GLP-1 and PYY. Participants were offered this optional procedure until the required number (*n* = 42) was obtained. The preload challenge protocol was completed at 1230 h. 

VAS is the recommended gold standard to measure self-reported appetite sensations [41,42]. At HNU, VAS was administered in a paper-and-pen format. The VAS consisted questions assessing Hunger, Fullness, Thoughts of Food (TOF), and Satisfaction, as detailed in our previous publication [43]. Briefly, the VAS is a 100 mm horizontal scale with two extreme feelings anchored at opposite ends of the scale. Participants responded by marking a vertical line on the 100 mm horizontal scale to best represent their current feelings of appetite. Participants completed the VAS independently without influence from other participants and investigators. The experimental setting and methodology for VAS assessment at the HNU adhered to the international guidelines for appetite studies [41,42], also as previously described [43].

### 2.8. Randomisation and Blinding

Eligible participants were randomised into one of the four dietary groups (HPLC, HPNC, NPLC, and NPLC) in a balanced ratio using an online randomisation tool [44]. Neither participants, nor the investigators were blinded to the allocated intervention. However, to minimise intervention bias, minimal interaction was ensured between participants in different dietary groups to avoid sharing information.

### 2.9. Statistical Methods

The sample size was determined based on the difference in BW loss between two independent groups. Based on similar BW data from Soenen’s study [10], 32 participants were required in each intervention group to detect a minimum difference of 2.8 kg in mean BW loss with 80% power and 95% significance level. Therefore, with 4 intervention groups in this study, a total of 128 participants were required to be enrolled in the study. With a 10% estimated drop-out rate, we tried to recruit 140 eligible participants. 

Descriptive data were reported as mean ± standard deviation (SD), and efficacy data as estimated marginal mean ± standard error of mean (SEM), unless otherwise stated. Participant characteristics at baseline were compared between dietary groups using one-way ANOVA. Continuous variables were checked for normal distribution with log-transformation applied if required. The completion rate between dietary groups was compared using Chi-squared test of independence.

The differential effects of dietary protein and CHO on the change in BW and its associated clinical outcomes were tested using a linear mixed model, by including a protein group (HP vs. NP); a CHO group (LC vs. NC); the interaction between protein and CHO groups (HPLC, HPNC, NPLC vs. NPNC); the Week and their interactions as fixed effects; participant as random effect; and baseline (Week 0) as a covariate. For appetite outcomes, Time (min) was additionally included as a fixed effect. Where there was a significant effect of Week or interaction between dietary groups and Week, a within-group or between-group post-hoc pairwise comparison was performed with Bonferroni’s correction. Area Under the Curve (AUC_0–210_) and incremental Area Under the Curve (iAUC_0–210_) were also calculated for appetite VAS following the trapezoid method using GraphPad Prism Version 9.3.0 (GraphPad Software, San Diego, CA, USA). Analyses of efficacy data adhered to the intention-to-treat (ITT) principle. Due to the nature of a longitudinal weight-loss intervention, missing values occurred when participants dropped out from the study. Missing values for BW, other anthropometry measurements, DXA-assessed body composition, and blood biomarkers were imputed using a multiple imputation (MI) by a chained equation. The procedure predicted these missing values based on dietary groups, age, ethnicity, anthropometry, and blood biomarkers using a random forest algorithm. The procedure generated an imputed dataset with 50 iterations, to create 5 imputed datasets, which were pooled for the final statistical output. To perform a sensitivity analysis, BW was additionally analysed by fitting observed cases in a linear mixed model using a restricted maximum likelihood approach. In contrast, self-reported outcomes (dietary intake and VAS) were analysed as observed cases only due to a greater error introduced by self-reporting. R (version 4.0.2; R Foundation for Statistical Computing, Vienna, Austria) was used to perform multiple imputations and its downstream statistical procedures. IBM Statistical Package for the Social Sciences (SPSS) software (version 25; IBM Corp., Armonk, NY, USA) was used to perform other statistical analyses. Statistical significance was set at *p* < 0.05. Values that lie > 3 × inter-quartile range from the mean were defined as extreme outliers.

## 3. Results

### 3.1. Participants

Of 121 female participants randomised, 93 completed the intervention, representing an overall completion rate of 77% (Figure 2). 79%, 83%, 87%, and 63% of participants randomised to the HPLC, HPNC, NPLC and NPNC group respectively completed the study. Chi-squared test of independence showed completion rate was not significantly related to dietary groups (X^2^ (3) = 5.6, *p* = 0.133). Two participants voluntarily withdrew from the study due to a seizure (serious adverse event) and a strained leg (adverse event). Mean age, BMI, anthropometry, and body composition, including FM and FFM were similar between 4 dietary groups at baseline Week 0 (Table 2).

### 3.2. Dietary Intake

Figure 3 summarised the intake of participants who completed the dietary records (Week 0, *n* = 116 out of 121 (96%); Week 4, *n* = 94 out of 110 (85%); Week 8, *n* = 87 out of 93 (94%)). The missing dietary records were not submitted by participants. At baseline, energy and macronutrient intake of participants were well-matched. When comparing the baseline intake to the AMDR, participants were consuming less CHO (mean = 40–43%en; AMDR = 45–65%en), but more fat (mean = 35–39%en; AMDR = 20–35%en). Of the 121 participants in the full cohort, 114 participants (94%) were allocated to the 3.9 MJ energy plan, 7 participants (6%) were allocated to the 4.6 MJ energy plan based on the estimated daily ER. Due to the small number of participants allocated to the 4.6 MJ energy plan, analysis was conducted as a single group irrespective of energy plan.

During the LED_pdr_ intervention, mean EI was less than half the pre-intervention baseline. Despite allowing the participants to regulate their variable meal intake based on appetite cues, reported mean daily EI did not differ significantly between dietary groups at either Week 4 or Week 8 (Protein×CHO×Week, *p* = 0.744) (Figure 3A). Protein intake was significantly higher in HP compared to NP at Week 4 and Week 8 (Protein×Week, *p* < 0.05, Figure 3B,C). However, at Week 8, the protein intake when expressed as g/kg BW was only trending towards higher in HPNC compared to NPNC (HPNC vs. NPNC, *p* = 0.080) (Figure 3B). Conversely, CHO intake was significantly lower in LC than NC at both Week 4 and Week 8 (CHO×Week, *p* < 0.001, Figure 3D). 

Both dietary records and urine urea-estimated protein intake showed the difference in protein intake between HP and NP to be lower than stipulated in the trial Protocol. Of the participants who completed both dietary record and concurrent 24-h urine collection (Week 0, *n* = 91 out of 121 (75%); Week 4, *n* = 73 out of 110 (68%); Week 8, *n* = 66 out of 93 (71%)), Bland–Altman analysis showed that 20% reliably reported their protein intake, while 40% under-reported, and 40% over-reported protein intake. Similarly, dietary records also showed the difference in CHO intake between LC and NC was smaller than the trial Protocol. 

### 3.3. Body Weight

BW loss was analysed both as observed cases and following imputation (Table 3). Both models showed all diets significantly decreased BW (Week, *p* < 0.001, both), and that the macronutrient composition of the LED_pdr_ had no significant effect on BW loss (Protein×Week, *p* > 0.05, both; CHO×Week, *p* > 0.05, both; Protein×CHO×Week, *p* > 0.05, both), in agreement with lack of significance in reported EI between diets. When analysed as observed cases, mean (±SEM) BW loss at Week 8 was 7.8 ± 0.2 kg (Week 8 vs. Week 0, *p* < 0.001), equivalent to 8.2% decrease from baseline. When grouped by diet, mean (±SEM) BW loss at Week 8 was 8.3 ± 0.5 kg, 6.6 ± 0.5 kg, 8.3 ± 0.5 kg, and 7.9 ± 0.5 kg for HPLC, HPNC, NPLC, and NPNC, respectively (Week 8 vs. Week 0, *p* < 0.001, all). When analysed following multiple imputation, mean (±SEM) BW loss at Week 8 was 7.5 ± 0.7 kg (Week 8 vs. Week 0, *p* < 0.001), equivalent to 7.9% decrease from baseline. When grouped by diet, mean (±SEM) BW loss at Week 8 was 8.4 ± 1.0 kg, 6.3 ± 1.0 kg, 8.5 ± 1.0 kg, and 8.6 ± 0.9 kg for HPLC, HPNC, NPLC, and NPNC, respectively (Week 8 vs. Week 0, *p* < 0.001, all).

### 3.4. Anthropometry and Body Composition

The LED_pdr_ intervention significantly decreased BMI, waist circumference, WHR, SBP, DBP, and FM (Week, *p* < 0.05, all) (Table 4), without a significant difference between dietary groups (Protein×Week, *p* > 0.05, all; CHO×Week, *p* > 0.05, all; Protein×CHO×Week, *p* > 0.05, all) (Appendix A). The mean (±SEM) decrease in FM in kg and percentage total BW at Week 8 was 5.7 ± 0.5 kg and 2.7 ± 0.5%, respectively (Week 8 vs. Week 0, *p* < 0.001, both), whereas the mean (±SEM) decrease in FFM was 1.4 ± 0.7 kg, trending towards significance (Week 8 vs. Week 0, *p* = 0.054). HP neither promoted the decrease in FM, nor retention of FFM over 8 weeks (Protein×Week, *p* > 0.05, both) (Appendix A). Of the total BW loss, approximately 80% was attributed to FM and 20% attributed FFM.

### 3.5. Serum Lipids

The LED_pdr_ intervention significantly decreased serum Total-C, HDL-C, LDL-C, and triglycerides, and increased serum NEFA (Week, *p* < 0.05, all) (Table 4), without a significant difference between dietary groups (Protein×Week, *p* > 0.05, all; CHO×Week, *p* > 0.05, all; Protein×CHO×Week, *p* > 0.05, all) (Appendix A). Despite the unfavourable decrease in serum HDL-C, the concomitant decrease in Total-C did not worsen the Total: HDL-C ratio (Week, *p* = 0.203). As expected, there was a significant increase in serum NEFA at Week 8 (Week, *p* < 0.001), likely indicating the mobilisation of fat stores due to negative energy balance.

### 3.6. Plasma Glucose and Glucoregulatory Peptides

The LED_pdr_ intervention significantly decreased plasma glucose, insulin and GIP (Week, *p* < 0.05, all) (Table 4), without significant difference between dietary groups (Protein×Week, *p* > 0.05, all; CHO×Week, *p* > 0.05, all; Protein×CHO×Week, *p* > 0.05, all) (Appendix A). As expected, the LED_pdr_ intervention significantly decreased plasma glucagon (Week, *p* < 0.001), but surprisingly NP significantly decreased plasma glucagon more than HP (Protein×Week, *p* = 0.050) (Appendix A).

### 3.7. Plasma Appetite-Related Gut Peptides

The LED_pdr_ intervention showed a trend towards a decrease plasma GLP-1 (Week, *p* = 0.051) (Table 4), without a significant difference between dietary groups (Protein×Week, *p* = 0.176; CHO×Week, *p* = 0.675; Protein×CHO×Week, *p* = 0.600) (Appendix A). At the end of the LED intervention (Week 8), plasma concentration of GLP-1 was 27.4 (95% CI: 6.3, 48.6) pg/mL lower than Week 0 (Week 8 vs. Week 0, *p* = 0.020). Conversely, the LED_pdr_ intervention did not significantly change plasma concentration of PYY (Week, *p* = 0.113) (Table 4), and there was no significant difference between dietary groups (Protein×Week, *p* = 0.806; CHO×Week, *p* = 0.963; Protein×CHO×Week, *p* = 0.785) (Appendix A).

### 3.8. Plasma Amino Acids

The LED_pdr_ intervention significantly decreased plasma concentration of 13 AAs (Week, *p* < 0.05, all) (Table 4), specifically 3 EAAs (phenylalanine, methionine, and threonine), 6 NEAAs (aspartic acid, asparagine, glutamic acid, alanine, tyrosine, and proline), and 4 NPAAs (hydroxyproline, taurine, citrulline, and ornithine). In contrast, serine was the only measured AA, which increased following the LED_pdr_ intervention (Week, *p* < 0.001). While glycine was significantly higher at Week 4 in comparison to Week 0 (Week 4 vs. Week 0, *p* < 0.001), this returned to baseline at the end of the LED_pdr_ intervention (Week 8 vs. Week 0, *p* = 1.000). Given the absence of significant change in dietary intake between Week 4 and Week 8, the trajectory of glycine was unexpected. Notably, hydroxyproline, glycine and glutamic acid were most susceptible to the effect of LED_pdr_ intervention when express as a percentage change, equivalent to a 23% decrease, a 22% increase and decrease, and a 22% decrease, respectively. BCAAs were not significantly changed by the LED_pdr_ intervention. The macronutrient composition of LED_pdr_ had no significant effect on most measured AAs, except an HP-driven decrease in taurine (Protein×Week, *p* = 0.045) (Appendix A).

### 3.9. Appetite Responses

Appetite responses were analysed as observed cases only (Week 0, *n* = 121; Week 8, *n* = 91) (Figure 4). When comparing the fasted appetite responses (t = 0 min), between Week 0 and Week 8 as all dietary groups combined, the LED_pdr_ intervention significantly increased mean (±SEM) Hunger by 9.2 ± 2.6 mm (Week, *p* < 0.001) and decreased mean (± SEM) Fullness by 4.9 ± 2.2 mm (Week, *p* = 0.027), but there was no significant change in TOF or Satisfaction (Week, *p* > 0.05, both). Although the multi-factorial analysis showed that macronutrient composition of LED_pdr_ did not significantly influence fasted appetite responses (Protein×Week, *p* > 0.05, all; CHO×Week, *p* > 0.05, all; Protein×CHO×Week, *p* > 0.05, all), it is noteworthy that the mean (±SEM) fasted Hunger rating at Week 8 following HPNC was significantly higher than Week 0 (15.9 ± 5.3 mm, paired *t*-test, *p* = 0.003) (Figure 4A).

When postprandial appetite was assessed in response to a standardised fixed meal as all dietary groups were combined, the LED_pdr_ intervention significantly suppressed postprandial Hunger and TOF, and significantly increased postprandial Fullness and Satisfaction, after adjusting for the fasted appetite responses (Week, *p* < 0.001, all). Importantly, the 4 dietary groups differentially suppressed postprandial Hunger (Protein×CHO×Week, *p* = 0.006) and increased postprandial Fullness (Protein×CHO×Week, *p* = 0.020). Post-hoc analysis revealed that postprandial Hunger suppression was significant following HPLC, HPNC, and NPLC (post-hoc, *p* ≤ 0.001, all), but not NPNC (post-hoc, *p* = 0.976). It is noteworthy that the significant suppression of Hunger following HPNC was a consequence of elevated fasted Hunger at Week 8, as evident by the loss of significance when analysed without adjustment for fasted Hunger ratings (post-hoc, *p* = 0.443). Postprandial Fullness was significantly increased following HPLC, HPNC, and NPLC (post-hoc, *p* < 0.01, all), but not NPNC (post-hoc, *p* = 0.194). The 4 dietary groups did not differentially suppress postprandial TOF (Protein×CHO×Week, *p* = 0.162) and increased Satisfaction (Protein×CHO×Week, *p* = 0.220). Both HP and NP similarly suppressed postprandial TOF (Protein×Week, *p* = 0.117) and increased postprandial Satisfaction (Protein×Week, *p* = 0.687). However, LC resulted in a greater suppression of postprandial TOF (CHO×Week, *p* < 0.001), and a greater increase in postprandial Satisfaction (CHO×Week, *p* = 0.045). AUC_0–210_ and iAUC_0–210_ appetite VAS were presented in Appendix A.

### 3.10. Postprandial Appetite-Related Gut Peptides Responses

In the subset of *n* = 42 participants (HPLC, *n* = 11; HPNC, *n* = 10; NPLC, *n* = 11; NPNC, *n* = 10) where postprandial blood samples were collected, *n* = 32 participants completed the study (HPLC, *n* = 9; HPNC, *n* = 10; NPLC, *n* = 9; NPNC, *n* = 4).

When analysed as all dietary groups combined, the appetite VAS responses in the subset were similar to the full cohort. Postprandial GLP-1 and PYY concentrations were not significantly different between Week 0 and Week 8, after they were adjusted for fasted concentrations (Figure 5, GLP-1, Week, *p* = 0.090; PYY, Week, *p* = 0.613). AUC_0–210_ and iAUC_0–210_ peptides were presented in Appendix A. Multi-factorial analysis was not conducted as the proportion of completers was imbalanced between the dietary groups.

## 4. Discussion

This study investigated the role of higher protein and lower CHO diets in promoting successful BW loss and the associated improvements in metabolic outcomes, with a particular focus on the modification of appetite responses based on macronutrient composition. The higher protein and lower CHO diets were delivered within the framework of an LED_pdr_. As expected, the LED_pdr_ intervention significantly lowered EI, resulting in significant LED-induced BW loss. Notably, despite this significant BW loss, no differential effect of higher protein or lower CHO was observed in this cohort of young and middle-age women with obesity, contrary to our original hypothesis. Although the final sample size was lower than the calculated sample size (Section 2.9), the difference in BW loss between any two dietary groups are much lower than the expected 2.8 kg based on the concordance in results obtained from the multiple imputation technique and the observed case analysis. Therefore, it appears that achieving the target sample size would not be likely to change our main findings. Furthermore, LC, but not HP, was identified as the primary component of the LED_pdr_ that resulted in the suppression of postprandial Hunger and TOF, and an increase in postprandial Fullness and Satisfaction in response to a standardised meal. The LC-induced improvement in postprandial appetite regulation as assessed in the laboratory did not translate into a detectable decrease in daily EI under free-living conditions when compared to NC, and hence did not result in differential BW loss. 

In agreement with the EFSA’s2012 report [11], increased feelings of satiety must be accompanied by a reduction in EI to induce a negative energy balance and promote BW loss. Rather than EI regulation, we observed some evidence of protein intake self-regulation during the 8-week intervention. The HPNC achieved the targeted protein intake at Week 4 (1.20 ± 0.23 g/kg BW), but was short of the target at Week 8 (1.08 ± 0.23 g/kg BW). The protein intake of NP at baseline was already higher than the target (NPLC = 1.01 ± 0.33 g/kg BW; NPNC = 0.99 ± 0.24 g/kg BW), hence decreasing protein intake to achieve 0.8 g/kg BW was likely challenging. Nevertheless, the macronutrient composition of LED had no significant differential effect on circulating the concentrations of most biomarkers, except for glucagon and taurine, which were differentially affected by the protein content of the LED_pdr_. Hence, improvement in markers of metabolic health was primarily a consequence of LED-induced BW loss, with very little differential response to macronutrient composition.

### 4.1. Effect of LED_pdr_ Intervention of BW, FM, and FFM

The 8-week LED_pdr_ intervention resulted in mean BW loss of 7.5 kg following multiple imputation and 7.8 kg following observed case analyses, equivalent to 7.9% and 8.2% BW loss from baseline, respectively. The BW loss achieved in our study was slightly lower than the 10 kg or 10% mean BW loss that has been reported in previous studies using an 8-week LED intervention with similar energy restriction [45,46,47,48]. Nevertheless, the BW loss achieved in our study was of clinical significance (>5% BW loss) and was accompanied by improvements in anthropometric measurements. Importantly, FM attributed to approximately 80% of the total BW loss, with a little loss in FFM, demonstrating that the standard LED (protein intake = 0.8 g/kg BW) can preserve FFM, in the absence of vigorous physical activity, in agreement to previous studies [17,49]. However, it is expected that an HP energy restricted diet when coupled with a component of exercise intervention could further promote the retention of FFM [50]. Contrary to our hypothesis, we did not observe significant differences in BW and FM loss between the HP and NP LED_pdr_. Although several meta-analyses have demonstrated the effect of HP diets on BW, FM and FFM loss [16,51,52,53], supporting findings from Soenen and colleagues [10], it is notable that the difference in BW, FM and FFM loss between HP and NP were modest in the meta-analysis. For instance, Vogtschmidt et al. [53] highlighted that HP promoted BW loss by 0.64 kg and FM loss by 0.55 kg, but had no significant effect on FFM, when compared to NP diets in randomised controlled trials of 4–156 weeks, aiming for BW loss and maintenance. Therefore, it is not entirely surprising that many individual weight-loss studies could not detect a modest difference in BW loss outcomes between HP and NP. In our study, the LED_pdr_ was already successful in promoting a negative energy balance, and an HP diet rarely translates into meaningful BW loss without an apparent difference in EI.

### 4.2. Effect of LED_pdr_ Intervention on Appetite

Our LED_pdr_ intervention significantly increased Hunger and decreased Fullness during the fasted state, and significantly suppressed Hunger and increased Fullness postprandially, indicative of an increased sensitivity towards the lack of nutrients in the fasted state and the abundance of nutrients in the postprandial state. Notably, our study measured appetite at the end of dietary intervention, before re-feeding; hence it is reflective of the participants’ appetite during negative-energy balance. The direction of change in appetite response whilst in negative energy balance, before re-feeding, is inconsistent in the LED/VLED literature. The DIOGENES sub-study reported no change in fasting Hunger, but the AUC Hunger decreased by 18% following LED intervention [54]. Following VLED interventions, Lyngstad et al. [55] reported an increase in fasting Hunger and a decrease in postprandial Hunger, similar to our findings; Adam et al. [56], Sumithran et al. [57], and Nymo et al. [58] reported no change in both fasting and postprandial Hunger; whereas Halliday et al. [59] reported a decrease in fasting Hunger but an increase in postprandial Hunger. One likely explanation for the difference in appetite responses is the state of ketosis during the negative energy balance. Several studies [57,58,60] have pointed out that an unfavorable increase in postprandial Hunger was absent when appetite was measured during the state of ketosis. Although an increase in Hunger is commonly cited as a physiological adaptation against BW loss [61,62], it is clear that there is an opportunity for intervention to promote satiety during negative energy balance [63].

Importantly, the macronutrient composition during the LED_pdr_ intervention differentially modified the postprandial appetite responses. Notably, the standardised meal during the preload challenge was identical for all participants at pre- and post-intervention. Therefore, the difference in postprandial appetite responses after consuming a standardised meal was likely related to the difference in macronutrient composition of LED_pdr_. Our study demonstrated that LC consistently led to favourable postprandial appetite responses, although the LC did not meet the criteria of a ketogenic diet (<50 g/day CHO) commonly known to suppress appetite [22]. Surprisingly, recent evidence revealed that ketosis can occur in individuals with obesity (baseline mean BMI = 35 kg/m^2^) following a 4.1 MJ LED with CHO intake up to 130 g/day [64], a level comparable to our NC. Although the CHO intake in our NC group is in the ‘normal’ range in comparison to the LC group, it is still much lower than the baseline CHO intake when expressed as grams. Since our present study did not measure ketone bodies, such as plasma beta-hydroxybutyrate and acetoacetate, we could not confirm the state of ketosis in these participants. However, it was possible that LC had a higher concentration of ketone bodies than the NC, resulting in a greater suppression of postprandial Hunger.

Whilst we observed an increased in postprandial satiety following the LED_pdr_ intervention when analysed as all dietary groups combined, there were no significant changes in the postprandial concentrations of GLP-1 and PYY between Week 0 and Week 8 in a subset of participants with similar appetite VAS responses. Due to the imbalanced proportion of completers in the subset of participants involved in repeated blood samplings, multi-factorial analysis was susceptible to attrition bias low statistical power; hence, it was not conducted. Although our present study showed that dietary groups did not differentially change fasted concentrations of GLP-1 and PYY, it remains speculative whether the increased postprandial satiety observed following LC may be related to increased postprandial concentrations of GLP-1 and PYY. Based on our previous review [28], we do not expect postprandial GLP-1 and PYY to be significantly and dramatically different between dietary groups, or to reach the ‘threshold’ required to affect appetite responses. Nevertheless, a detailed analysis of postprandial metabolic response following the standardised meal is currently underway in our laboratory, aiming to understand the longitudinal association between the change in appetite-related biomarkers and appetite responses following the 8-week LED_pdr_ intervention using regression techniques similar to Martins et al. study [65].

### 4.3. Effect of LED_pdr_ Intervention on Biomarkers of Metabolic Health

In this cohort of women with obesity and with normal to slightly elevated glycaemia, the LED intervention significantly decreased fasted glucose concentration. While a prior study has shown that HP energy-restricted diet eliminated weight loss-induced improvements in insulin sensitivity [66], the larger multinational PREVIEW study showed that the use of LED followed by a 3-year HP or NP weight maintenance diet significantly reduced the risk of developing type-2 diabetes in individuals with pre-diabetes [67]. This potential role of protein in promoting and reducing insulin resistance has been discussed in detail by Rietman et al. [68], with a potential gap in our understanding, warranting further research. Of interest is also the role of HP LED in ameliorating the decrease in fasted concentration of glucagon, which may translate into an increase in gluconeogenesis when compared to NP, a process hypothesised to increase energy expenditure and suppressing appetite [69]. Our study did not result in a meaningful improvement in lipid profile due to the concurrent decrease in HDL-C along with Total-C. This observation is expected following an LED [46], due to the low dietary fat intake [70]. Furthermore, obesity also features an underlying altered AA metabolism, especially higher fasted concentration of BCAAs and aromatic AAs, but lower glycine and serine [71,72,73,74,75] Of the 23 AAs measured in our study, concentrations of 13 AAs decreased, while serine increased. Unexpectedly, the concentration of glycine was higher at Week 4 but returned to baseline at Week 8. The decrease in fasted concentration of AAs is likely related to an improvement in AA catabolism, and is unrelated to the level of dietary protein intake during the LED_pdr_. BCAAs are more frequently studied in the field of obesity and type-2 diabetes than the other AAs and is often reported to decrease following dietary BW loss interventions [66,76,77,78]; however, this was not observed in our study. We similarly also did not observe a decrease in BCAAs following a total diet replacement LED (Cambridge Weight Plan^®^) in the PREVIEW-NZ sub-study [79]. The dietary AA composition, rather than the total protein intake, may underlie the differences in AA metabolism [34]. In general, a total diet replacement LED program is required to deliver at least 75 g at Protein Digestibility-Corrected Amino Acid Score of 1.0 [80], which is expected to be different from food-based interventions. Overall, 8-week changes in metabolic profile are predominantly related to BW loss under negative energy balance conditions, rather than a macronutrient composition of LED_pdr_.

### 4.4. Strengths and Limitations

The strength of this study lies in the ability to elucidate the differential role of HP versus LC diet in BW loss and appetite using a multifactorial approach. Additionally, to allow for the satiating properties of macronutrients to regulate appetite and therefore EI, our study design allowed participants to regulate their EI based on appetite cues; hence, BW loss was not obscured by a fixed EI. Nevertheless, there were several limitations. First, the effect of macronutrients on appetite was studied using an energy-restricted LED_pdr_, hence could not be generalised to an *ad lib* non-energy restricted diet. Increasing protein intake from RDI to 1.2 g/kg BW may not be able to induce voluntary decrease in EI when prescribed EI within the LED is already low. However, in the context of a non-energy restricted diet, HP has shown to result in a lower EI when compared to NP [24,81,82]. Second, there is no reliable tool by which to assess EI in free-living participants, since self-reported dietary records are prone to reporting bias and error [83]. Although a 24-h urine sample was collected, its completeness was not assessed via p-aminobenzoic acid (PABA) tests [84]. The possibility of incomplete urine collection and inaccuracies in urine-estimated protein intake, could not be ruled out. Third, the difference in protein and CHO intake between groups were smaller than anticipated. Hence, any potential effect of consuming 1.2 g/kg BW protein on BW loss and its associated outcomes when compared to protein intake at the RDI level might have been weakened. Fourth, plasma ketone bodies were not measured; therefore, we could not confirm the state of ketosis in these participants. Lastly, postprandial blood samples were only available in a subset of participants, and the significant loss of participants in the NPNC subset hindered our ability to conduct multi-factorial analysis for postprandial peptide outcomes.

## 5. Conclusions

During the 8-week LED_pdr_, all dietary groups achieved statistically significant and clinically meaningful BW loss (mean = 7.5 kg) in this cohort of women aged 18–60 years with obesity. However, HP (mean reported intake = 1.08–1.20 g/kg BW) did not further promote BW loss compared to NP (mean reported intake = 0.91–0.99 g/kg BW), independent of the CHO content of LED_pdr_. Although HP did not further promote the retention of FFM in comparison to NP, the protein content of both the HP and NP arms exceeded the RDI, with the reported difference in protein intake between HP and NP also lower than anticipated. The success of an LED_pdr_ in driving a substantial negative energy balance may have also precluded the modest effect of HP in promoting energy deficit in comparison to NP. Therefore, mechanisms that underlie protein-induced BW loss, such as EI suppression and retention of FFM, were not detectable in this study. Unexpectedly, the LC but not the HP component of the LED_pdr_ significantly suppressed postprandial Hunger and increased postprandial Fullness in response to a standardised meal. Due to the smaller-than-anticipated difference in protein intake between HP and NP, any potential yet modest beneficial effect of an HP diet on satiety, BW loss, and FFM retention might have been precluded by the success of the LED_pdr_.

## Figures and Tables

**Figure 1 nutrients-14-00538-f001:**
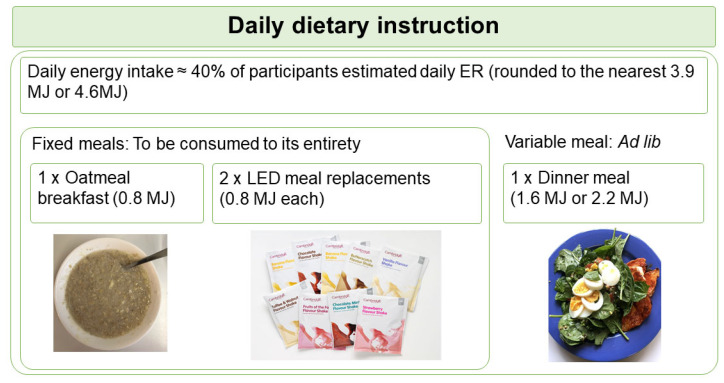
Schematic shows the daily dietary instruction given to participants. Participants were instructed to consume 1 oatmeal breakfast, 2 LED meal replacements, and 1 variable meal daily in a “1 + 2 + 1” format. Participants were instructed to consume the variable meal to appetite (*ad lib*). ER, energy requirement; LED, Low Energy Diet.

**Figure 2 nutrients-14-00538-f002:**
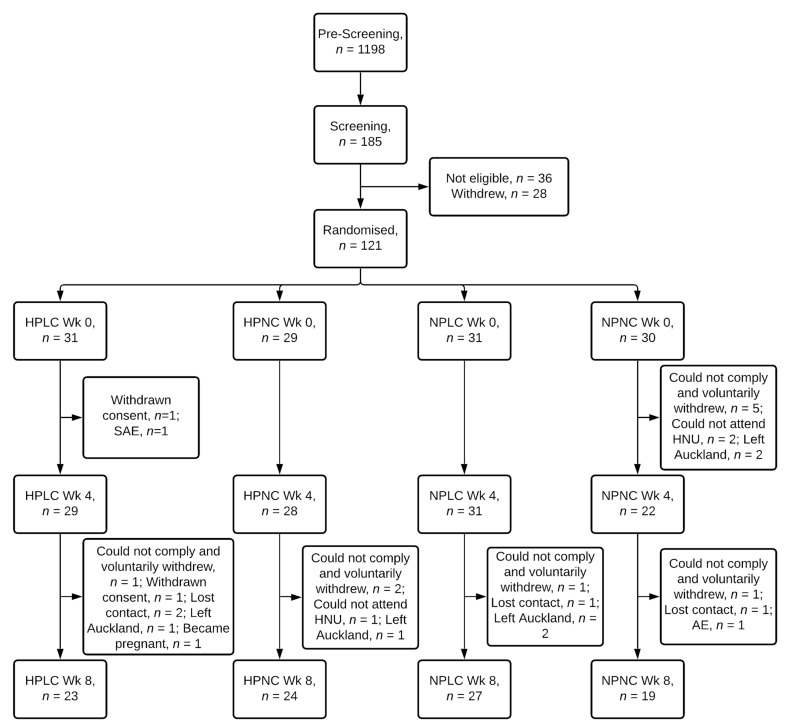
Flow diagram of participants. HPLC, Higher Protein Lower Carbohydrate; HPNC, Higher Protein Normal Carbohydrate; NPLC, Normal Protein Lower Carbohydrate; NPNC, Normal Protein Normal Carbohydrate; Wk, Week; SAE, serious adverse event; AE, adverse event.

**Figure 3 nutrients-14-00538-f003:**
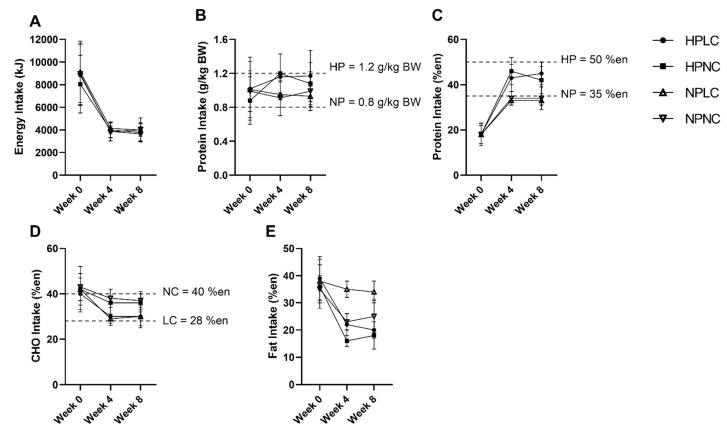
Observed mean (±SD) (**A**) energy (en%), (**B**) protein (k/kg BW), (**C**) protein (%en), (**D**) CHO (en%), and (**E**) fat (en%) intake of participants after excluding extreme outliers (*n* = 2 from HPNC at Week 4, *n* = 1 from HPLC and *n* = 2 from HPNC at Week 8). Number of observations at Week 0: HPLC, *n* = 30; HPNC, *n* = 29; NPLC, *n* = 30; NPNC, *n* = 27. Number of observations at Week 4: HPLC, *n* = 25; HPNC, *n* = 21; NPLC, *n* = 26; NPNC, *n* = 20. Number of observations at Week 8: HPLC, *n* = 23; HPNC, *n* = 19; NPLC, *n* = 26; NPNC, *n* = 17. BW, body weight.

**Figure 4 nutrients-14-00538-f004:**
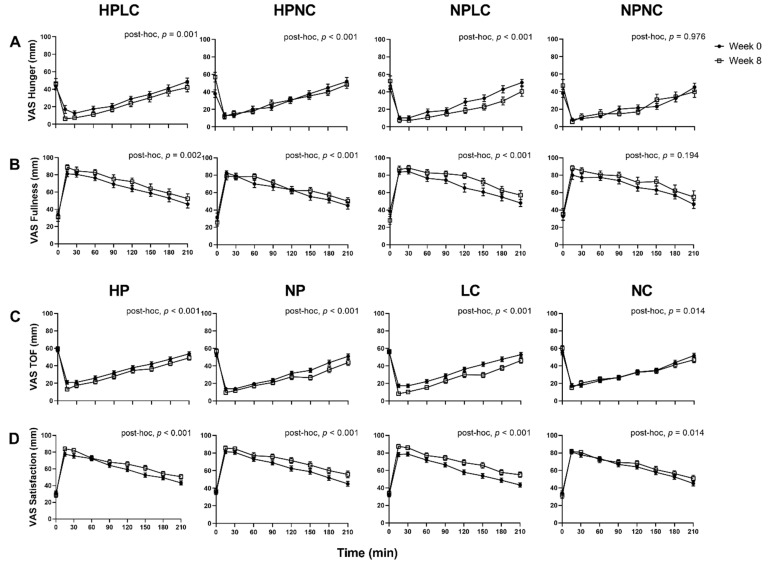
Observed mean (±SEM) subjective feelings of appetite. (**A**) Hunger and (**B**) Fullness grouped by HPLC, HPNC, NPLC, and NPNC; and (**C**) TOF and (**D**) Satisfaction grouped by HP, NP, LC, and NC. Number of participants completed appetite VAS at Week 0 and Week 8 were *n* = 121 and *n* = 91, respectively. Observed cases data were analysed using a linear mixed model, adjusted for differences in fasted ratings (t = 0 min). Post-hoc analysis shows the statistically significant difference in postprandial appetite responses between Week 0 and Week 8.

**Figure 5 nutrients-14-00538-f005:**
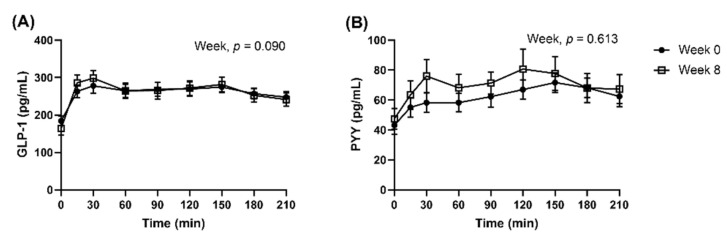
Observed mean (±SEM) concentrations of (**A**) GLP-1 and (**B**) PYY presented as all dietary groups combined. Number of participants completed the repeated blood samplings procedure at Week 0 and Week 8 were *n* = 42 and *n* = 32, respectively. Observed cases data were analysed using a linear mixed model, adjusted for differences in fasted concentrations (t = 0 min).

**Table 1 nutrients-14-00538-t001:** Planned macronutrient composition of each treatment.

	Higher Protein (HP)		Normal Protein (NP)	
	Lower CHO (LC)	Normal CHO (NC)	Lower CHO (LC)	Normal CHO (NC)
Protein	50%en	50%en	35%en	35%en
CHO	28%en	40%en	28%en	40%en
Fat	22%en	10%en	38%en	25%en

%en, percentage of total energy.

**Table 2 nutrients-14-00538-t002:** General characteristics of participants at baseline (Week 0).

Characteristics	HPLC (*n* = 31)	HPNC (*n* = 29)	NPLC (*n* = 31)	NPNC (*n* = 30)	*p*-Value
Age (years)	39.0 ± 10.1	37.4 ± 12.0	41.2 ± 10.2	41.3 ± 12.4	0.472
BW (kg)	95.5 ± 11.3	95.6 ± 12.4	93.9 ± 12.8	95.3 ± 15.8	0.956
Height (m)	1.63 ± 0.07	1.64 ± 0.07	1.63 ± 0.08	1.64 ± 0.07	0.863
BMI (kg/m^2^)	35.9 ± 4.0	35.3 ± 3.5	35.3 ± 4.0	35.2 ± 4.4	0.898
Waist circumference (cm)	100.7 ± 7.5	101.6 ± 8.9	100.5 ± 8.8	100.1 ± 10.1	0.935
Hip circumference (cm)	120.8 ± 9.4	121.3 ± 8.8	119.5 ± 7.4	118.5 ± 10.8	0.627
WHR	0.84 ± 0.07	0.84 ± 0.04	0.84 ± 0.06	0.84 ± 0.06	0.910
SBP (mmHg)	119 ± 13	115 ± 13	115 ± 13	116 ± 13	0.684
DBP (mmHg)	65 ± 6 ^a^	62 ± 6 ^a,b^	61 ± 6 ^b^	64 ± 7 ^a,b^	0.040
FM (kg)	44.6 ± 7.8	44.5 ± 8.4	44.0 ± 7.5	44.6 ± 10.6	0.990
Percentage FM (%)	46.8 ± 4.7	46.6 ± 4.0	47.0 ± 2.9	46.7 ± 4.5	0.981
FFM (kg)	50.3 ± 6.2	50.5 ± 5.9	49.3 ± 6.2	50.2 ± 6.9	0.884

Data are presented as mean (±SD). Differences in baseline measurements between dietary groups were compared using analysed using One-Way ANOVA. Participant characteristics at baseline were well-matched (ANOVA, *p* > 0.05, all), except for DBP (ANOVA, *p* = 0.040). Significant between-group differences were indicated using different letters in the superscript. BW, body weight; BMI, body mass index; WHR, waist-hip ratio; SBP, systolic blood pressure; DBP, diastolic blood pressure; FM, fat mass; FFM, fat-free mass.

**Table 3 nutrients-14-00538-t003:** The effect of dietary groups on body weight (BW) (kg) measured at Week 0, 4 and 8.

Analysis Methods	Week	HPLC	HPNC	NPLC	NPNC	*p*-Value		
						Protein×Week	CHO×Week	Protein×CHO×Week
Multiple Imputation ^1^	0	95.1 ± 0.7	95.1 ± 0.7	95.0 ± 0.7	95.1 ± 0.7	0.733	0.454	0.928
	4	89.5 ± 0.8	90.9 ± 0.8	89.6 ± 0.7	90.4 ± 1.0			
	8	87.2 ± 1.3	88.8 ± 0.9	86.6 ± 0.8	87.6 ± 1.2			
Observed Cases ^2^	0	94.0 ± 1.8	96.3 ± 2.4	93.5 ± 2.5	94.1 ± 3.7	0.432	0.116	0.469
	4	88.7 ± 1.8	91.5 ± 2.6	88.0 ± 2.2	89.0 ± 3.7			
	8	85.7 ± 2.0	89.6 ± 2.8	85.2 ± 2.2	86.1 ± 3.8			

^1^ BW data reported as estimated marginal means ± SEM following linear mixed model, adjusted for baseline BW. ^2^ BW data reported as observed mean ± SEM (Week 0, *n* = 121; Week 4, *n* = 110; Week 8, *n* = 93). *p*-value was reported following linear mixed model after statistically adjusted for baseline BW.

**Table 4 nutrients-14-00538-t004:** The effect of treatment diets on anthropometry, body composition, and circulating metabolites measured at Week 0, 4 and 8.

Variables	Baseline	Change from Week 0		*p*-Value
	Week 0	Week 4	Week 8	Week
Anthropometry				
BMI (kg/m^2^)	35.4 ± 0.1	−1.8 ± 0.2 ***	−2.8 ± 0.2 ***	**<0.001**
Waist (cm)	100.7 ± 0.4	−4.0 ± 0.6 ***	−7.1 ± 0.7 ***	**<0.001**
WHR	0.840 ± 0.004	−0.003 ± 0.005 ^ns^	−0.019 ± 0.008 ^ns^	**0.038**
SBP (mm Hg)	116.3 ± 0.7	−4.0 ± 1.0 ***	−7.4 ± 1.0 ***	**<0.001**
DBP (mm Hg)	62.7 ± 0.4	−2.6 ± 0.5 ***	−3.4 ± 0.5 ***	**<0.001**
Body composition				
FM (kg)	44.4 ± 0.2	NA	−5.7 ± 0.5 ***	**<0.001**
FM (% body mass)	46.8 ± 0.3	NA	−2.7 ± 0.5 ***	**<0.001**
FFM (kg)	50.1 ± 0.4	NA	−1.4 ± 0.7 ^ns^	0.065
Serum lipids				
Total-C (mM)	5.29 ± 0.06	NA	−0.91 ± 0.11 ***	**<0.001**
HDL-C (mM)	1.44 ± 0.02	NA	−0.21 ± 0.03 ***	**<0.001**
Total:HDL-C (mM)	3.95 ± 0.10	NA	−0.22 ± 0.15 ^ns^	0.203
LDL-C (mM)	3.45 ± 0.05	NA	−0.61 ± 0.08 ***	**<0.001**
Trig (mM)	1.52 ± 0.08	NA	−0.54 ± 0.12 ***	**0.001**
NEFA (mM)	0.49 ± 0.02	NA	0.17 ± 0.03 ***	**<0.001**
Glucose and glucoregulatory peptides				
Glucose (mM)	5.67 ± 0.05	−0.31 ± 0.07 ***	−0.36 ± 0.08 ***	**<0.001**
Insulin (pg/mL)	724.2 (CI: 646.1, 812.1)	−231.1 (CI: −334.6, −127.9) ***	−185.3 (CI: −284.3, −86.2) ***	**<0.001**
Glucagon (pg/mL)	49.0 (CI: 45.6, 53.0)	−7.4 (CI: −12.4, −2.5) **	−10.3 (CI: −15.9, −4.7) ***	**<0.001**
GIP (pg/mL)	45.6 (CI: 41.7–49.9)	−7.9 (CI: −14.1, −1.7) *	−6.9 (CI: −13.1, −0.8) ^ns^	**<0.001**
Gut peptides				
GLP-1 (pg/mL)	172.9 (CI: 158.5, 188.6)	−7.2 (CI: −26.1, 11.8) ^ns^	−27.4 (CI: −48.6, −6.3) *	0.051
PYY (pg/mL)	37.6 (CI: 32.7, 43.3)	−5.4 (CI: −10.5, −0.2) ^ns^	−2.8 (CI: −7.6, 2.1) ^ns^	0.113
Branched-chain amino acids				
Leucine (µM)	121.0 ± 1.4	−1.3 ± 2.4 ^ns^	1.4 ± 2.5 ^ns^	0.515
Isoleucine (µM)	65.1 ± 1.2	1.2 ± 1.8 ^ns^	1.4 ± 1.7 ^ns^	0.701
Valine (µM)	236.3 ± 3.8	−1.4 ± 5.0 ^ns^	2.9 ± 4.7 ^ns^	0.645
Other essential amino acids				
Phenylalanine (µM)	55.8 ± 0.7	−2.5 ± 1.0 *	−3.2 ± 1.2 *	**0.025**
Methionine (µM)	27.9 ± 0.5	−1.6 ± 0.6 *	−2.2 ± 0.7 **	**0.006**
Lysine (µM)	83.8 ± 1.2	0.4 ± 1.8 ^ns^	−3.3 ± 1.7 ^ns^	0.086
Histidine (µM)	53.9 ± 1.2	−1.7 ± 1.7 ^ns^	−0.4 ± 1.8 ^ns^	0.597
Threonine (µM)	111.7 ± 1.9	−1.6 ± 2.6 ^ns^	−8.6 ± 2.8 **	**0.003**
Tryptophan (µM)	39.7 ± 0.7	−1.7 ± 1.1 ^ns^	−3.4 ± 1.4 *	0.053
Non-essential amino acids				
Glycine (µM)	275.5 ± 6.1	60.2 ± 9.1 ***	−8.0 ± 9.3 ^ns^	**<0.001**
Aspartic acid (µM)	5.5 ± 0.3	−1.6 ± 0.4 ***	−1.4 ± 0.4 **	**<0.001**
Asparagine (µM)	48.0 ± 0.6	−0.1 ± 0.9 ^ns^	−2.4 ± 1.0 *	**0.016**
Glutamic acid (µM)	55.0 ± 1.8	−12.5 ± 2.7 ***	−12.1 ± 2.8 ***	**<0.001**
Glutamine (µM)	534.7 ± 5.0	−5.2 ± 7.2 ^ns^	−9.9 ± 7.5 ^ns^	0.458
Arginine (µM)	76.2 ± 1.7	−0.4 ± 2.6 ^ns^	−1.9 ± 2.5 ^ns^	0.738
Alanine (µM)	404.6 ± 5.3	−59.5 ± 7.8 ***	−70.2 ± 7.8 ***	**<0.001**
Serine (µM)	105.5 ± 1.7	10.7 ± 2.4 ***	14.6 ± 2.6 ***	**<0.001**
Tyrosine (µM)	66.9 ± 0.9	−7.6 ± 1.2 ***	−8.9 ± 1.4 ***	**<0.001**
Proline (µM)	208.1 ± 3.6	−17.5 ± 5.3 **	−39.0 ± 5.8 ***	**<0.001**
Non-proteogenic amino acids				
Hydroxyproline (µM)	13.8 ± 0.5	−1.8 ± 0.7 *	−3.2 ± 0.7 ***	**<0.001**
Taurine (µM)	114.7 ± 4.7	1.2 ± 7.1 ^ns^	−15.9 ± 6.9 ^ns^	**0.033**
Citrulline (µM)	29.4 ± 0.3	−2.3 ± 0.5 ***	−2.1 ± 0.5 ***	**<0.001**
Ornithine (µM)	38.9 ± 0.9	−3.8 ± 1.3 *	−2.7 ± 1.2 ^ns^	**0.010**

Data reported as estimated marginal means ± SEM or 95% CI following linear mixed model using multiple imputed data. BMI, body mass index; WHR, waist-hip ratio; SBP, systolic blood pressure; DBP, diastolic blood pressure; FM, fat mass; FFM, fat-free mass; Total-C, total cholesterol; HDL-C, high-density lipoprotein cholesterol; Total: HDL-C, Total-to-HDL cholesterol ratio; LDL-C, low-density lipoprotein cholesterol; Trig, triglycerides; NEFA, non-esterified fatty acids; GIP, gastric inhibitory polypeptide; GLP-1, glucagon-like peptide-1; PYY, peptide YY. Asterisks marked a significant different from Week 0. * *p* < 0.05, ** *p* < 0.01, *** *p* < 0.001. ns, not significantly different than Week 0 (*p* > 0.05). Bold represent a statistically significant main ANOVA effect of Week (*p* < 0.05).

## Data Availability

De-identified data may be shared and made available upon reasonable request to the corresponding author and subject to an approved proposal and data access agreement.

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
