# Peer review of "Does a Higher Protein Diet Promote Satiety and Weight Loss Independent of Carbohydrate Content? An 8-Week Low-Energy Diet (LED) Intervention"

_nutrients, 2022, doi:10.3390/nu14030538_

Round 1
Reviewer 1 Report
Very well designed and presented study.
1.- It surprised me how you calculated very well the recruitment number and the extra participants to account for drop outs, but finally you didn't reach the number targeted, even if you were very near. I don't think it would have a affected your conclusions but with the high number o preparticipants why do you think that happened?
2.- Your result demonstrated that the low diet intake is the key for weight loss independently of the amount of proteins, and that HP did not promote the retention of FFM in comparison to NP, because in both arms the content needed of proteins was largely surpassed. High protein intake has shown to prevent loss of muscle mass (Cava et al. 2007; Backx et al. 2016; ), but has adverse effects on glucose metabolism (Smith et al 2016) eliminating the induced improvement in muscle insulin sensitivity and in that way other reports suggested that high protein intakes are involved in the pathogenesis of insulin resistance.
On the other hand, we can mantain FFM without protein intake. Martin-Rincon et al (2019) and Calbet et al. (2014) revealed that lean muscle mass was prevented with long low intensity resistance excercise that may have increased your body weight loss results, even in very high energy deficit (5000 kcal), comparing two very high caloric restrictions, one arm with just protein intake and the other just with carbs. Truth is that their intervention lasted only 4 days but in those days weight loss was significant and FFM preserved in both arms.
3.- Did you measured leptin as a satiating signal? It is demonstrated that it is upregulated with severe energy déficit but also with high protein intakes.
4.- As you addressed in your study other arms could have been interesting to include in your protocol. It can be interesting to see the effects of a normal to hyper caloric diet even with the same low/ high carb, and low/ high protein intakes.
Author Response
Very well designed and presented study.
1.- It surprised me how you calculated very well the recruitment number and the extra participants to account for drop outs, but finally you didn't reach the number targeted, even if you were very near. I don't think it would have a affected your conclusions but with the high number o preparticipants why do you think that happened?
Author response 1:
We appreciate the reviewer for commending our calculation. Based on our calculation, in order to detect a significant 2.8 kg difference in BW loss between any two dietary groups we required a total of n = 128 participants in the study (Lines 229 – 233). As with many longitudinal intervention studies, we had to take into consideration the number of participants that we predict will ‘drop out’, i.e. not complete the study. Therefore, the goal is to recruit a total of n = 140 participants (Line 234). Following screening of n = 185 prospective participants, we identified n = 149 eligible participants, but 28 of the eligible participants decided to withdraw before the commencement of the study, resulting in n = 121 participants in our intention-to-treat (ITT) population (Figure 2).
Prior to the commencement of the study, we encouraged eligible participants to discuss participation with friends and family members, to ensure additional support during the 8-week LED intervention to keep them motivated. To further promote compliance during the study, we followed standardised protocols used at the Human Nutrition Unit, alongside regular support and guidance from our research team. The nature of the LED intervention weight-loss program means that it can be difficult to undertake, requiring substantial changes to an individual’s lifestyle, which often precludes commitment to the study, which was what we experienced.
Of the n = 121 participants in our intention-to-treat (ITT) population that were randomised to the 4 dietary groups we had n = 28 participants who withdrew (Figure 2), resulting in n = 93 completers. Whilst our sample size calculation showed that n = 128 participants was desirable, we achieved n = 121 ITT population and n = 93 completers population. Using the statistical advice of consultant and co-author Mr Daniel Barnett, we undertook a sensitivity analysis to determine whether the drop-outs were likely to have an important effect on the outcomes of the trial. To do this, we used a multiple imputation technique to firstly impute missing data to assess the effect of dietary groups on BW loss in the ITT population. We then secondly also further assessed the effect of dietary groups on BW loss using an observed-case analysis. The results from both statistical methods were in agreement, both showing that the difference in BW loss between any two dietary groups were lower than the 2.8 kg (Lines 329 and 334). Therefore, we inferred that reaching the target sample size of n = 128 was not likely to change the main finding of the study. In response to the reviewer question, we have now expanded and included this in the Discussion text (Lines 445 – 450).
2.- Your result demonstrated that the low diet intake is the key for weight loss independently of the amount of proteins, and that HP did not promote the retention of FFM in comparison to NP, because in both arms the content needed of proteins was largely surpassed. High protein intake has shown to prevent loss of muscle mass (Cava et al. 2007; Backx et al. 2016; ), but has adverse effects on glucose metabolism (Smith et al 2016) eliminating the induced improvement in muscle insulin sensitivity and in that way other reports suggested that high protein intakes are involved in the pathogenesis of insulin resistance.
On the other hand, we can mantain FFM without protein intake. Martin-Rincon et al (2019) and Calbet et al. (2014) revealed that lean muscle mass was prevented with long low intensity resistance excercise that may have increased your body weight loss results, even in very high energy deficit (5000 kcal), comparing two very high caloric restrictions, one arm with just protein intake and the other just with carbs. Truth is that their intervention lasted only 4 days but in those days weight loss was significant and FFM preserved in both arms.
Author response 2:
We thank the reviewer for their detailed comments and insights regarding the potential adverse effect of HP diets in the pathogenesis of insulin resistance. Rietman et al. (2014) initially questioned whether HP intake reduces or elicits insulin resistance. They then proposed that HP intake may be associated with insulin resistance but in the context of non-energy restricted diets, conversely HP diets have been shown to promote weight loss during energy-restricted diet. The contrary finding that HP energy-restricted diet blunted the weight-loss induced improvement in muscle insulin sensitivity as described by Smith et al. (2016) is indeed interesting. Large multinational diabetes prevention studies such as the PREVIEW study (Raben et al. 2020), did not find significant differences between HP vs NP weight maintenance diets in terms of diabetes prevention in 2223 individuals with pre-diabetes. Therefore, there is still a significant gap in knowledge whether HP may have unfavourable effects on insulin sensitivity during negative, neutral, and positive energy balance. We have taken on board the reviewer’s comments and have now acknowledged the potential gap in our understanding of HP diets and their role in weight loss-induced improvements in insulin sensitivity in the Discussion (Lines 538 – 545).
We additionally appreciate the reviewer’s input on the importance of exercise on maintenance of lean muscle mass. We acknowledge that HP energy-restricted diets when paired with exercise intervention may prevent the loss of fat-free mass in longer-term studies as has been reported in the detailed systematic review by Weinheimer et al. in 2010 which we have referenced in text Lines 481 – 482.
3.- Did you measured leptin as a satiating signal? It is demonstrated that it is upregulated with severe energy déficit but also with high protein intakes.
Author response 3:
We are in agreement with the reviewer regarding leptin’s possible role in satiety, albeit that data in this area is not conclusive. We did not measure leptin in this study as our a priori hypothesis was driven towards the peptides GLP-1 and PYY. While energy restriction has been shown to decrease circulating leptin concentration (Klempel and Varadi, 2011), and HP non-energy restricted diets have been shown to result in lower daily ad lib energy intake with decreased circulating leptin concentrations (Weigle, 2004), whether HP vs NP diets, administered under the same level of energy deficit, may differentially change circulating leptin concentration is an interesting research question worth investigating in future studies.
Furthermore, progression towards obesity is associated with leptin resistance (Izquierdo et al. 2019). Despite a reduction in leptin concentration following weight loss, leptin sensitivity may be increased. Future research aiming to investigate the role of leptin in regulation of satiety should consider assessment of leptin sensitivity in addition to leptin concentration per se.
4.- As you addressed in your study other arms could have been interesting to include in your protocol. It can be interesting to see the effects of a normal to hyper caloric diet even with the same low/ high carb, and low/ high protein intakes.
Author response 4:
Yes, we agree with the reviewer that it would be interesting to assess the effect of these diets at different states of energy balance (negative, neutral, and positive energy balance) when BW loss is not the primary endpoint. In epidemiology studies, we frequently observe that the reduction in CHO intake is associated with an increase in protein and fat intake, which in turn has been shown to increase the risk of type-2 diabetes. However, acute single-meal studies and weight-loss studies have shown that HP or LC is beneficial for enhanced satiety, insulin action, and body composition. To shed light on the role of HP or LC diets on daily appetite regulation and insulin action, we consider an investigation of HP or LC diets at neutral and positive energy balance would be beneficial.
Reviewer 2 Report
The paper is well written and requires minor clarification on some of the methods that the authors have selected
It's interesting that the subjective feeling of appetite was measured using VAS, how exactly was this done? Other studies had showed that participants are quite bad at reporting their subjective feelings and requires multiple other scales to ensure consistency. Can the authors justfiy why VAS is used here?
Sample size assumption feels correct, but in reality a pilot should've been carried out. Just to be clear, did Soenen also looked at protein intake and BW? Why is this referenced here?
It's unclear why the authors used random forest algorithm here, was the data incomplete?
Table 3. Is this BW in kg? I would've assumed that it would be more meaningful to report it as % difference to baseline?
Fig 1. Did the authors considered calculating Area Under the Curve?
Discussion reads fine but would be nice to have some subheaders to walk the readers through the main key topic/results that the authors are trying to explain.
Author Response
The paper is well written and requires minor clarification on some of the methods that the authors have selected
It's interesting that the subjective feeling of appetite was measured using VAS, how exactly was this done? Other studies had showed that participants are quite bad at reporting their subjective feelings and requires multiple other scales to ensure consistency. Can the authors justfiy why VAS is used here?
Author Response 1:
We thank the reviewer for their comment with regard the use of VAS in the measurement of appetite, and agree that this is a method that is constantly under review in the field of appetite research. We have now elaborated further on how the measurement was conducted at the Human Nutrition Unit in the Methods section 2.7 (Lines 207 – 217). While VAS has its limitations, as acknowledged by the reviewer, it continues to remain the recommended gold standard according to the international guidelines for appetite studies to measure self-reported appetite sensations (Blundell et al 2010; Gibbons et al 2019), and hence used in our current study. Additionally, and importantly, the use of VAS to measure appetite also allowed standardised comparisons between our study with other previously published studies. We had an experienced research team using standardised protocols for conducting VAS assessments at the Human Nutrition Unit, and have utilised this methodology in multiple other appetite studies conducted in our lab (Poppitt et al. 2011, Wiessing et al. 2012, Poppitt et al. 2013, Poppitt et al. 2015, Poppitt et al. 2018, Lim et al., 2021). We have updated the references supporting use of the VAS methodology (Blundell et al. 2010; Wiessing et al. 2012; Gibbons et al 2019) in text Line 216 - 217.
Sample size assumption feels correct, but in reality a pilot should've been carried out. Just to be clear, did Soenen also looked at protein intake and BW? Why is this referenced here?
Author Response 2:
We note the reviewer’s suggestion that a pilot study could have further informed our calculations on sample size in this weight loss intervention. Our study built upon Soenen’s prior intervention conducted in The Netherlands (described Lines 74 – 77), and accordingly was powered utilising the expected difference in body weight loss, and variability in this data based upon the standard deviation reported in Soenen’s prior publication. The reviewer is correct in that Soenen et al. did also investigate the differential role of HP vs LC on low energy diet (LED)-driven weight loss using a multi-factorial approach (referenced in text Lines 59 – 61). The similarity in cohort, diet composition and intervention design were the key reasons for utilising data that derived from this Dutch study to obtain the sample size calculations for our current intervention.
It's unclear why the authors used random forest algorithm here, was the data incomplete?
Author Response 3:
The reviewer is correct in that, due to the nature of an LED weight loss intervention, missing data values did occur when participants dropped out from the study. For example, n = 11 participants withdrew between Week 0 and Week 4, and n = 17 participants withdrew between Week 4 and Week 8 (Figure 1). Therefore, the data was incomplete. This information has now been added in text Lines 252 – 253. Following statistical consultation with Mr Daniel Barnett, also co-author, the random forest algorithm is an unbiased efficient prediction algorithm that is used to handle missing values (Breiman, 2001), hence was used as part of the multiple imputation procedure for imputing missing values.
Table 3. Is this BW in kg? I would've assumed that it would be more meaningful to report it as % difference to baseline?
Author response 4:
Yes, BW data presented in Table 3 is in kg. Table 3 is intended to present the BW data estimated from the multiple-imputation, which was also baseline adjusted, against the body weight derived from the observed-case only. We wanted to show that there was no significant bias between results obtained from the two different statistical methods utilised, and that the estimated BW following multiple imputation was in concordance with that from the observed cases.
Importantly, we agree with the reviewer that % BW loss is also a meaningful way to report this data. Accordingly, we reported the 8-week % BW loss in text in the Results section 3.2 Line 329 following observed-case analysis and again in Line 333 following multiple-imputation analysis. Since there were no between-group differences in BW loss, the % BW loss by groups at each week have not been presented.
Fig 1. Did the authors considered calculating Area Under the Curve?
Author response 5:
We thank the reviewer for this comment and can confirm that we did calculate Area Under the Curve (AUC) and incremental AUC (iAUC) values, which we have now made clear and expanded in Section 2.9 Lines 248 – 251. The Figures for AUC and iAUC appetite VAS outcomes have been provided as Supplementary Information rather than within the main document in order to meet publication limits by the Journal. Alongside the Repeated Measures plots are the AUCs and iAUCs results to allow comparison. We also would like to acknowledge that we had mistakenly labelled Figure 4, as Figure 1 which we have now amended on Page 14, Line 432.
While repeated measures analysis is the most robust method to analyse appetite VAS according to the international guidelines for appetite studies (referenced in text Line 215 (Blundell et al., 2010)), additionally presenting AUC and iAUC VAS allows further additional comparisons between other published studies which report AUC and iAUC data.
We note that statistical significance differs when appetite VAS is expressed as AUC and iAUC. Sometimes, AUC and iAUCs may help with the interpretation of repeated measures analysis, or vice versa. The effect of HPNC diet on Hunger VAS is a good example. Although HPNC had a significantly lower iAUC0-210 Hunger at Week 8 relative to Week 0, the AUC0-210 Hunger at Week 8 was similar to Week 0 (Figure S1(B)). Based on the repeated measures plot, the greater suppression of iAUC0-210 Hunger was due to the higher fasted Hunger (t = 0 min) at Week 8, as the Hunger ratings between t = 15 – 210 min were similar. Consequently, despite the greater suppression of Hunger in response to a standardised preload meal at Week 8 following HPNC, the higher Hunger rating at t = 0 min abolished the difference in Hunger when expressed as AUC0-210. This observation is reported in text Lines 419 – 422, and is supported by the additional AUC0-210 and iAUC0-210 presented as Supplemental Material (in text Lines 429 – 430).
Discussion reads fine but would be nice to have some subheaders to walk the readers through the main key topic/results that the authors are trying to explain.
Author response 6:
We note the reviewers request and agree that this does make the Discussion much easier to follow. We have now added subheaders within the text at Lines 469, 496, 536, and 573.